# Toward Core-Free Pavement Compaction Evaluation: An Innovative Method Relating Asphalt Permittivity to Density

**Kyle Hoegh** [1],*  , **Roger Roberts** [2], **Shongtao Dai** [1] **and Eyoab Zegeye Teshale** [1]

1   Minnesota Department of Transportation, Materials and Road Research, Maplewood, MN 55109, USA
2   Geophysical Survey Systems Inc, Nashua, NH 03060-3075, USA
*   Correspondence: kyle.hoegh@state.mn.us; Tel.: +1-1651-366-5526

**Abstract:** Asphalt pavement compaction quality control and quality assurance (QC/QA) are traditionally based on destructive drilled cores and/or nuclear gauge results, which both are spot measurements representing significantly less than 1 percent of the in-service pavement. Ground penetrating radar (GPR) is emerging as a tool that can be used for nondestructive continuous assessment of asphalt pavement compaction quality through measuring the pavement dielectric constant. Previous studies have established that asphalt pavement dielectric constant measurements are inversely proportional to the air void content for a given asphalt mixture. However, field cores are currently required to calibrate the measured dielectric constant to the pavement density. In this paper, a method is proposed to eliminate the need for field calibration cores by measuring the dielectric constant of asphalt specimens compacted to various air void contents. This can be accomplished with a superpave gyratory compactor (SGC), which is routinely used in the pavement industry to fabricate 6 in. (15.2 cm.) diameter specimens. However, this poses difficulties with the GPR antenna height, direct coupling, and the Fresnel zone in relation to the asphalt specimen dimension limitation. These challenges are overcome by employing a plastic spacer with a known dielectric constant between the SGC specimen and the antenna. The purpose of the spacer is to reduce GPR wave speed so that the signal reflected from the specimen is separated from the direct coupling effects at an antenna height where the Fresnel zone of the GPR is not affected by the specimen dimension. The specimen dielectric constant can then be measured using the reflection coefficient-based surface reflection method (SR) or the pulse velocity-based time-of-flight method (TOF). Also, The Hoegh–Dai model (HD model) is demonstrated to reasonably predict pavement density based on the results of field measurements and corresponding core validation, especially as compared to the conventional exponential model. Results are presented from multiple days of paving on one project, as well as a single paving day on a project with significantly different mix properties. The agreement between the HD model, coreless prediction, and field cores shows the promise for implementation of dielectric-based asphalt compaction evaluation without the need for destructive field core calibration.

**Keywords:** GPR; dielectric constant; relative permittivity; relative density; electromagnetic wave; pavement; non-invasive; air voids; asphalt; QC/QA; ground penetrating radar; non-destructive testing; electromagnetic waves; signal processing; antennas and radar systems; geosciences

## 1. Introduction

Ground penetrating radar (GPR) is a non-destructive testing method that provides a continuous evaluation of pavement using electromagnetic waves [1,2]. The successful use of GPR for pavement assessments in addition to surface layer asphalt density [3–6] includes asphalt and base layer depth

profiling [7], pavement material assessment [8], and detection of standing water or drainage issues [9]. The degree of asphalt concrete (AC) pavement compaction is a critical factor affecting both the air void content in roadway structures and the proper functioning of pavements [10–14]. Estimates show that each 1 percent increase in air void content above 7 percent air voids results in approximately 10% loss in pavement life [10,11].

Despite the importance of air void content, the state-of-the-practice compaction evaluation is both destructive and point source-based, with quality control and quality assurance (QA/QC) assessments based on significantly less than 1% of the placed pavement. To address this limitation, ground penetrating radar (GPR) has been used for decades to measure the surface dielectric constant (also referred to as the dielectric) of asphalt pavement using non-contact horn antennas typically mounted on vehicles [15], or other innovative methods including step-frequency and array-based systems [16–19]. Since the measured dielectric constant is a function of all the components in the asphalt mixture (i.e., aggregates, binder, and additives), the measured dielectric requires unique calibration factors for each specific mix-design to be converted to actual air void content. This dielectric-based compaction assessment assumes that changes in dielectric caused by mix variability are negligible compared to changes in air voids. Studies have shown this is generally a reasonable assumption [4–6]. Other factors like water content and pavement temperature have been considered. While the current field core calibration is based on measurements up to 200 degrees Fahrenheit, the proposed study relies on measurements conducted at room temperature. A laboratory study conducted by Iowa State suggests this discrepancy in temperature will not be a significant factor. The study showed that the permittivity (dielectric), measured in the temperature range of 80 to 200 degrees Fahrenheit using high-frequency impulses (10 GHz), was almost independent of temperature [20]. The dielectric constant measurement methods used in this study do not separate the effects of moisture in the analysis. Therefore, the presence of water should increase the measured dielectric and skew the predicted air void content lower. In this study, the asphalt specimen air void measurements were made using a vacuum sealing method [21] to keep the specimen dry prior to dielectric testing. Water is also present immediately after the final roller compactor is applied on the pavement. However, since the pavement is typically 180 degrees Fahrenheit (82 degrees Celsius), the water evaporates quickly. Therefore, care was taken to ensure water had evaporated from the pavement surface prior to conducting field testing, following protocol similar to previous studies [22,23].

More recently, smaller-size dipole-type antennas have been used to measure the dielectric of asphalt mixture to a higher degree of accuracy [22]. These antennas have been used in a non-contact manner, similar to the afore-mentioned horn antennas, to provide a continuous and comprehensive statistics-based evaluation of compaction quality as it relates to various construction activities and mix design strategies [23]. As mentioned above, in order to determine the asphalt pavement compaction density in the field with GPR, a relationship between the dielectric constant and air void content of the mixture needs to be established to convert measured dielectric constants to in-place pavement density. However, calibration of the dielectric to asphalt air void content has relied on destructive coring of the as built pavement to calibrate to each mix. This destructive and labor-intensive calibration method must be repeated each time a significant change in mix design occurs. Mechanistic mix-modeling approaches have been proposed [5,6], but they use input values such as aggregate and binder dielectric constant that are not readily measured [8]. Al-Qadi et. al. [8] demonstrated that the dielectric of the binder and aggregate components can be obtained by back calculation from density and dielectric constant data. However, this approach also relies on core-drilling for accurate dielectric to air void conversion. Further, proper calibration should include air void content on the extreme values, which are not always feasible to identify and collect in the field, and require sophisticated models to add stability for extrapolating outside of the bounds of collected data [23]. While field core-based dielectric to air void content conversion has been shown to be achievable, the destructive and labor-intensive nature have created a barrier to widespread implementation, especially in environments like Texas and Minnesota where source aggregates are not constant [22,23].

To address these issues, a novel method based on the use of asphalt mixture specimens manufactured using the superpave gyratory compactor (SGC) is proposed. Compacted asphalt specimens representing a range of air void contents observable in the field are then used to calibrate the dielectric constant to the specific production mix without the need for field cores. These 6-inch cylindrical samples are referred to as "asphalt specimens" or "specimens" herein. The GPR equipment that allows for input of the calibration curve to directly output density is referred to as the density profiling system (DPS) herein. The gyratory compactors used to fabricate the specimens are readily available, since the asphalt paving industry relies on them as part of typical asphalt mix design QA/QC procedures [24]. The proposed procedure is also suitable for mix design sensitivity analysis that may be used to determine the necessity and frequency of field core validations and recalibrations. An additional advantage of using the asphalt specimen-based calibration is that agencies typically have test summary sheets of production mixes with standard reporting mechanisms for any changes to the mix that are tracked during a project [24]. These reporting mechanisms could provide agencies with the necessary input to trigger recalibration, which would be useful for development of dielectric–mix sensitivity-based implementation specifications.

Since the Fresnel zone of the GPR antennas exceeds the diameter of the specimens used for calibration, it was found to be inaccurate to make measurements at the field recommended heights of 6 to 12 in. (15–30 cm). The measurements were affected by surface edge diffractions. Reducing the antenna to a height with an acceptable Fresnel zone (smaller than the specimen diameter) is not feasible, since the timing of the reflected signal from the asphalt specimen is within the range of the direct coupling noise of the system and multiple reflections between the surface and the antenna. Dielectric constant measurements of small asphalt specimens (6 in. diameter) can only be accurately measured using waveguide methods [25] or microwave free-space methods [26] that are impractical outside of laboratory conditions, require additional measurement equipment that may not represent the frequency characteristics of the field equipment, and require samples of known and regular dimensions that are not easily produced with typical asphalt production mix. The other option, increasing the diameter of the specimen used for calibration, is equally impractical since the SGC is the most widely used compactor for the fabrication of asphalt samples and it is limited to a maximum diameter of 6 in. Considering these limitations, the proposed method only requires the existing field dielectric constant measuring and asphalt specimen fabrication equipment, but addresses the small specimen issue by introducing spacer material with a known dielectric constant [27,28]. This material is shown to sufficiently slow down the wave, and to increase the time gap between the direct coupling and the asphalt specimen response without having to raise the antenna.

The proposed method does not require field cores, which damage the in-place pavement, to develop the calibration curve. The method is amenable to implementation in that the material is readily available, and equipment used to create the asphalt specimens is already part of the QC/QA process. The current testing and reporting of production mix characteristics can be used for similar reporting of mix changes that need to be tracked for routine DPS implementation. Thus, the proposed method is a modification of the field air launched method that resolves the aforementioned limitations, and allows for fully non-invasive conversion of asphalt dielectric to density without requiring any additional equipment or drastically changing the QA/QC reporting mechanisms.

## 2. Materials and Methods

Two basic procedures are required for the coreless calibration method including (1) the fabrication of asphalt specimens with a range of air void contents and (2) testing each asphalt specimen to calculate the dielectric constant. The former follows AASHTO T312 that is widely used by the pavement industry [24]. By following the AASHTO T312 procedure, a certified lab technician can fabricate asphalt specimens with targeted air void contents spanning the range of asphalt compaction air voids observable in the field. Testing of these standard specimens involves an innovative testing technique and associated calculation methodologies that are the subject of this paper. Two calculation

methodologies are discussed. One is called surface reflection (SR) method, which is based on measured surface reflection amplitude from one asphalt specimen surface; the other one is called time-of-flight (TOF) method, which is based on measured travel times from the top and bottom of the asphalt specimen. The procedure for estimation of both surface reflection and time-of-flight based dielectric (TOF dielectric) measurements of the asphalt specimen using the Delrin®spacer are given in Sections 2.2 and 2.3, respectively. The problem, Fresnel zone-related edge effects versus signal isolation, is introduced in the antenna characteristics section.

*2.1. Antenna Characteristics*

When measuring the dielectric constant of a specimen, a typical GPR waveform contains a direct wave from transmitter to receiver (direct coupling). In both of the proposed methods, sufficient isolation between direct coupling and the reflection from the surface of the asphalt specimen must be achieved. A 12 in. (30 cm) height above the pavement surface is recommended by the manufacturer during field survey measurements for sufficient isolation from direct coupling interference (see Figure 1). At this height, the ground footprint area (first Fresnel zone) of the GPR waveform can be estimated using Fr ~ 0.5 c $(t/f)^{1/2}$, where Fr is the first Fresnel zone radius of the footprint, c is speed of light, t is two-way travel time, and f is antenna frequency [29].

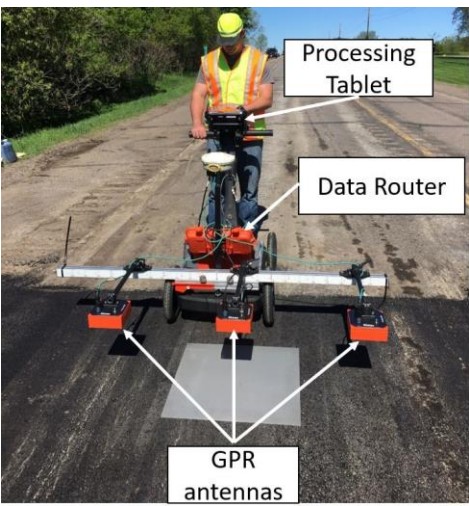

**Figure 1.** Density profiling system (DPS) equipment during data collection.

The estimated Fr is 3.8 in. (9.7 cm) at an antenna height of 12 in. (30 cm), while the radius of a gyratory specimen is typically 3 in (7.6 cm). This means that the area of a gyratory specimen (a reflector) is less than the area bordered by the circular zone with the radius (Fr). Since most reflected waveform energy comes from this zone, the measured reflected signal from gyratory specimens is affected by the geometry of the specimen (edge of the specimen) and does not accurately reflect material dielectric constant. Therefore, the recommended testing arrangement cannot be effectively employed when testing gyratory compacted asphalt specimens.

An experiment was carried out to verify the theoretical antenna footprint area, where a metal plate was placed initially approximately 2 ft (60 cm) away from an antenna at the standard 12 in. (30 cm) height (see Figure 2). The plate was gradually moved toward the center of the antenna while monitoring the dielectric constant output. It was noticed that the output dielectric constant suddenly increased when the edge of the metal plate was approximately 4in. (10 cm) away from the center of the antenna, which indicates that the antenna detected the metal plate at this radius (see Figure 3). The approximate 4 in. (10 cm) distance was determined by measuring the distance with a tape measure at the location corresponding to the dielectric magnitude at scan 780 where the increase in amplitude

occurred. This measured distance of 4 in. (10 cm) between the edge of the metal plate and the center of the antenna was in agreement with the calculated Fresnel zone radius of 3.8 in. (9.7 cm).

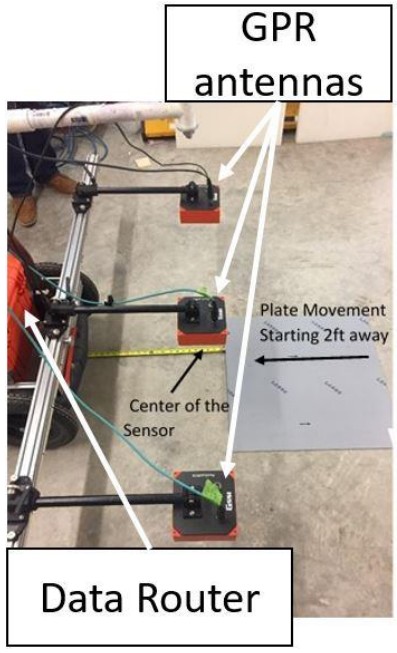

**Figure 2.** Picture of the DPS experiment showing presence of metal reflection at the first Fresnel zone.

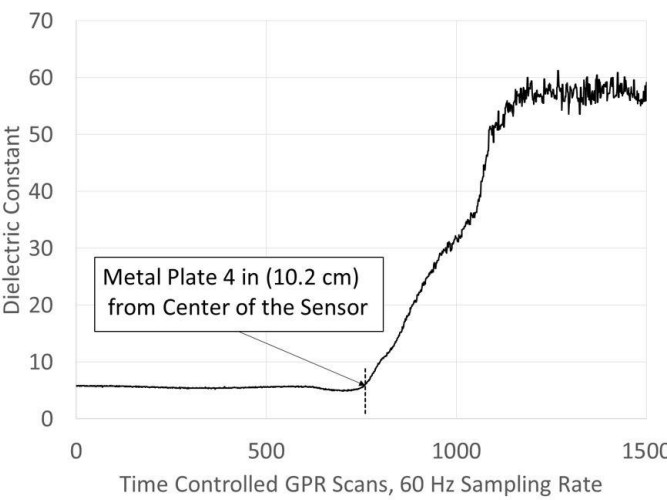

**Figure 3.** Experimental data confirming the first Fresnel zone approximation applies to the GPR antenna.

As discussed above, the direct coupling may affect the specimen top surface reflection if the antenna does not have enough height above the surface. Ideally, this direct coupling effect should be able to be eliminated from the waveforms of the specimen measurements by subtracting the GPR waveform from air measurement that only contains the direct coupling and antenna characteristics. However, it has been noticed that direct coupling does not always occur at the same location relative to the time axis on the GPR waveform. In other words, the amplitudes of the direct coupling from different air measurements do not occur at the same time (they show variation on the time axis). This phenomenon makes the subtraction difficult and incomplete. To characterize the noise that cannot be accounted for solely by subtracting the mean direct coupling, 1000 "air-scans" were collected and analyzed by looking at 2 standard deviations from the mean at each sample location. The air scans are conducted by ensuring there is nothing but air within 18 in. (46 cm) of the antenna during data

collection. In this arrangement, any recorded signal is not a result of any physical material but rather of interference intrinsic to the antenna that should be filtered (e.g., multiple reflections within the measurement equipment, direct coupling, etc.). To evaluate the response accurately within the direct coupling zone, the signal (mean response) was plotted along with the noise (represented by the 2 standard deviations from the mean) of the 1000 responses at each sample (see Figure 4). It can be observed that significant variability is observed earlier in time, during the direct coupling response, especially at locations between local maxima and minima where the slope of the signal is highest. This variability affects the precision of the asphalt specimen response, thus creating a need to separate the asphalt specimen response in time, away from the direct coupling. Analysis of the asphalt specimen response within the direct coupling time window can thus alter the reflection amplitude (which affects the surface reflection (SR) method) and shift the apparent arrival time (which affects the time-of-flight (TOF) method). The air launched method used in the field cannot be applied to asphalt specimens, since the asphalt specimen response at an antenna height that accounts for the antenna footprint is not sufficiently separated in time from the direct coupling noise shown in Figure 4.

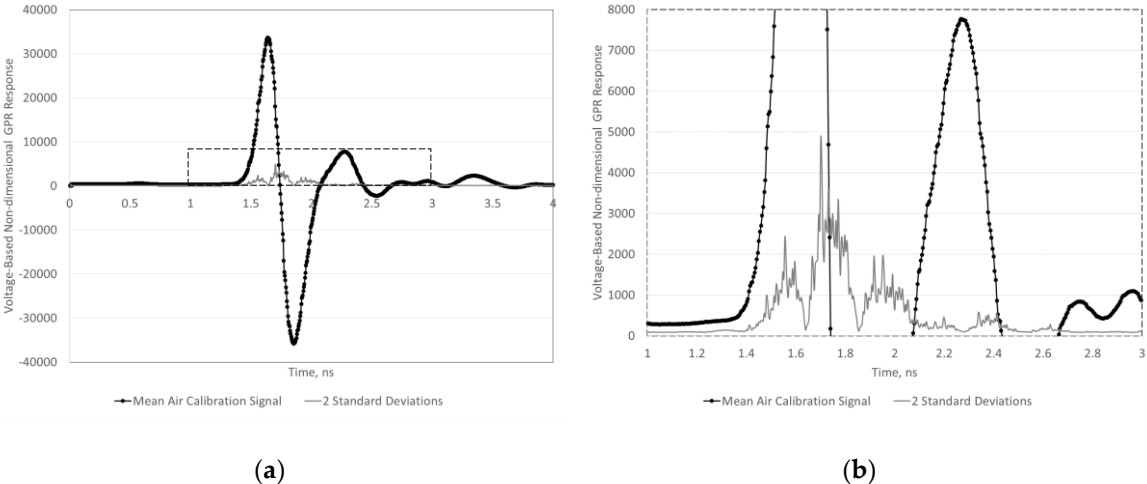

(**a**)                                                 (**b**)

**Figure 4.** Variability of the signal with respect to direct coupling shown by air calibration scan mean and 2 standard deviations: (**a**) zoomed out and (**b**) zoomed in on the section bordered with a gray dotted line in Figure 4a with a clearer view of the 2 standard deviation variations.

The basic concept, to overcome Fresnel zone height requirements and the corresponding direct coupling interference characterized above, involves replacing the air medium with a known dielectric medium that reduces the pulse velocity such that the asphalt specimen reflection occurs after the direct coupling noise. A solid spacer medium, such as the acetal homopolymer plastic (Delrin®), can be used since the asphalt sample can be measured in a stationary location, unlike continuous coverage field measurements.

## 2.2. Approach 1: Dielectric Obtained from the Surface Reflection

Using the proposed spacer method as the model, the secondary effects caused by the limited dimensions of the 6 in. (15 cm) asphalt specimen are discussed with respect to the surface reflection dielectric (SR dielectric) by evaluating the edge diffraction path in comparison to the surface reflection. Figure 5 shows the ray paths associated with the different reflections and diffractions.

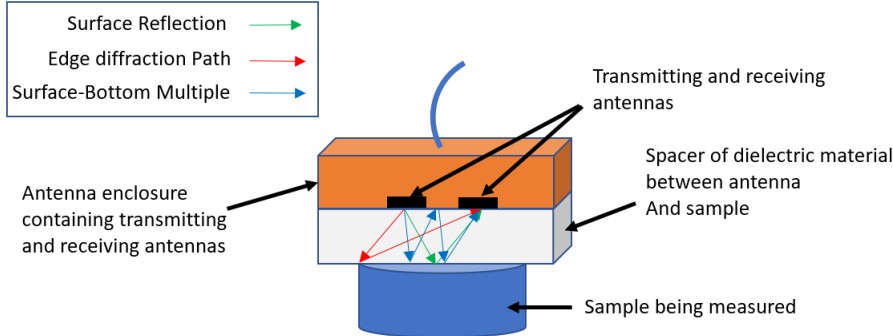

**Figure 5.** Cut-away view showing the ray paths indicating sources of major reflections and diffractions arriving from surface of sample being measured.

The surface reflection dielectric constant calculation (SR dielectric) is the closer analog to the field testing procedure of the two proposed dielectric calculation methods. This method is also the more straight-forward process to calculate the dielectric from the isolated surface reflection amplitude. It is possible to calculate the arrival times of the different reflections and diffractions and assess if there is a certain combination of spacer dielectric and thickness that provides a time window that ensures the reflection from the surface precedes the arrival of other reflected and diffracted energy given the following known information:

1. Spacing between the transmitting and receiving antennas;
2. Radiated pulse length;
3. Thickness of spacer material;
4. Dielectric of spacer material; and
5. Diameter of core or asphalt specimen being measured.

The method assumes the following:

6. Fresnel's plane wave propagation equations can be used to approximate the reflection amplitude from the surface;
7. The dielectric of the sample being tested is assumed to be relatively homogeneous across the surface of the material. This assumption might not be valid for asphalt samples containing large aggregate or samples containing a non-uniform aggregate distribution.

The separation distance between the transmitting and receiving parts of the antenna used in the study—a GSSI Model 42600 antenna—is 2.4 in. (6 cm). The functional separation of the antenna was determined experimentally to be 1.7 in. (4.4 cm) using one-way and two-way travel time calculations in a known-thickness, high-density polyethylene (HDPE) material with a known 2.3 frequency independent dielectric. The necessity for experimentally determined separation is caused by the impulse not exhibiting perfect point source behavior. The portion of the pulse length of interest is the time from the onset of the reflection pulse to its negative peak amplitude. The negative peak amplitude was chosen, since it occurs earlier in time and thus has less likelihood of being affected by the reflected waves from the asphalt specimen edge occurring later in time. Figure 6 shows an isolated reflection pulse and the time length of interest, which is approximately 0.34 nanoseconds.

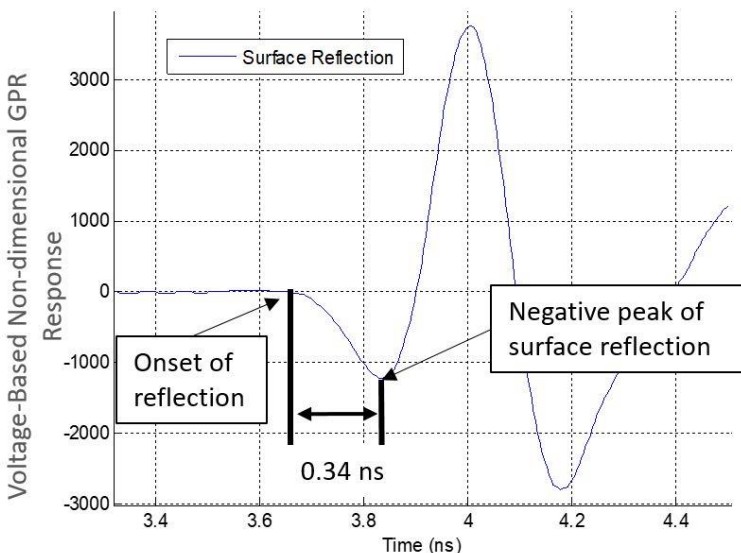

**Figure 6.** Isolated reflection from surface reflection indicating the time length of interest is about 0.34 ns and is measured from the onset of the reflection to the negative peak.

Using the known antenna-related parameters and a fixed diameter of 6 in. (15 cm) for the material to be measured, the arrival times of each reflection and diffraction can be calculated and plotted. For example, Figure 7 shows the arrival times of the reflections and diffractions for an air spacer between the antenna and the sample. It is shown that for an air spacer thickness up to 2.1 in (5.4 cm), the negative peak of the surface reflection arrives later than the beginning of both the surface multiple and the edge diffraction.

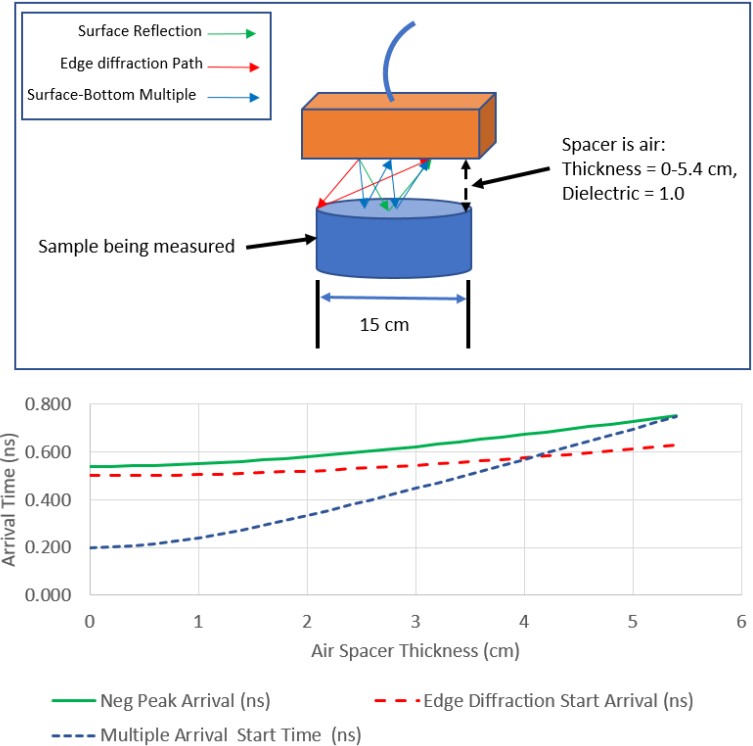

**Figure 7.** Calculations of the arrival time of the negative peak of the surface reflection versus the start of the edge diffraction and multiple reflection arrival times for an air spacer of varying thicknesses.

Analysis of the calculated arrival times of the different ray paths for spacers was conducted to determine the appropriate dimensions and dielectric constant of the spacer. Delrin®samples were selected as plastic with a comparatively high dielectric constant (Dielectric = 2.88) that is available in 6 in. by 6 in. (15.2 cm × 15.2 cm) sheets with various nominal thicknesses. Travel times of the first negative peak are plotted in Figure 8, along with multiple arrival and edge diffraction start times to characterize isolation of the negative peak. It can be observed that within a thickness range of 1.4 in. to 2 in. (3.8 cm to 5.1 cm), the negative peak arrival occurs prior to the edge diffraction and surface-bottom multiple. Based on this analysis, Delrin®spacers with 1.5 in. (3.8 cm) thickness were selected for use in this study. Since the overlap in arrival between the signal of interest (surface reflection) and the multiple arrival onset occurs at a lower thickness—1.4 in. (3.5 cm)—the Delrin spacer isolates the negative peak from interference.

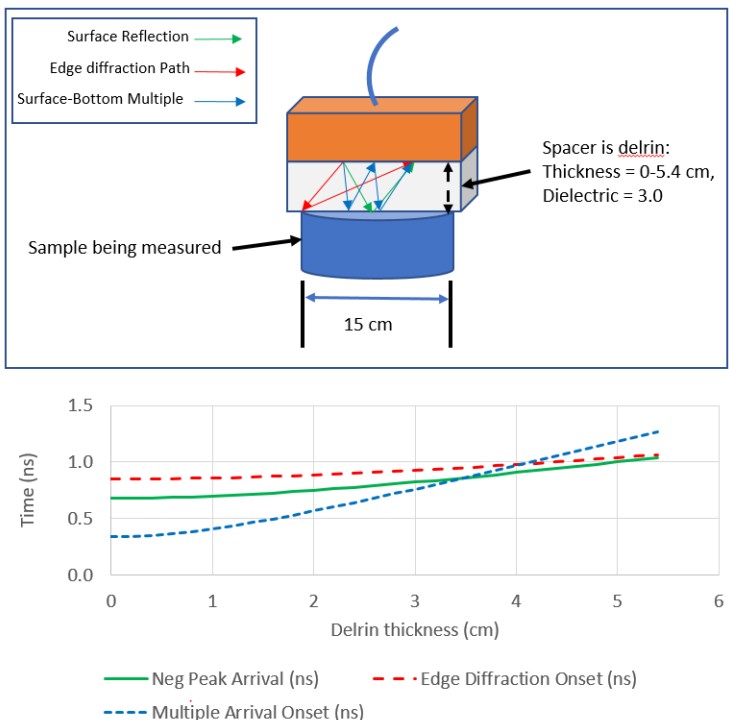

**Figure 8.** Calculations of the arrival time of the negative peak of the surface reflection versus the start of the edge diffraction and multiple reflection arrival times for 3.0 dielectric material, of varying thicknesses.

The surface reflection from the sample being measured is typically superimposed on clutter arising from various reflections and diffractions near to and inside the antenna enclosure. This clutter needs to be removed in order to isolate the amplitude of the surface reflection. For air-launched field tests, this is accomplished by averaging and subtracting multiple scans with only air within the time range of analysis. For the proposed Delrin®-launched test, this is accomplished by adding Delrin® to make a background measurement where Delrin® is the only material within the time range of analysis. A second measurement is performed with a metal plate placed underneath the spacer, analogous to the metal plate amplitude calculation in the air launched method [23]. The final measurement is made with the asphalt sample placed underneath the spacer. Figure 9 shows the set-up for the three measurements and the corresponding scans.

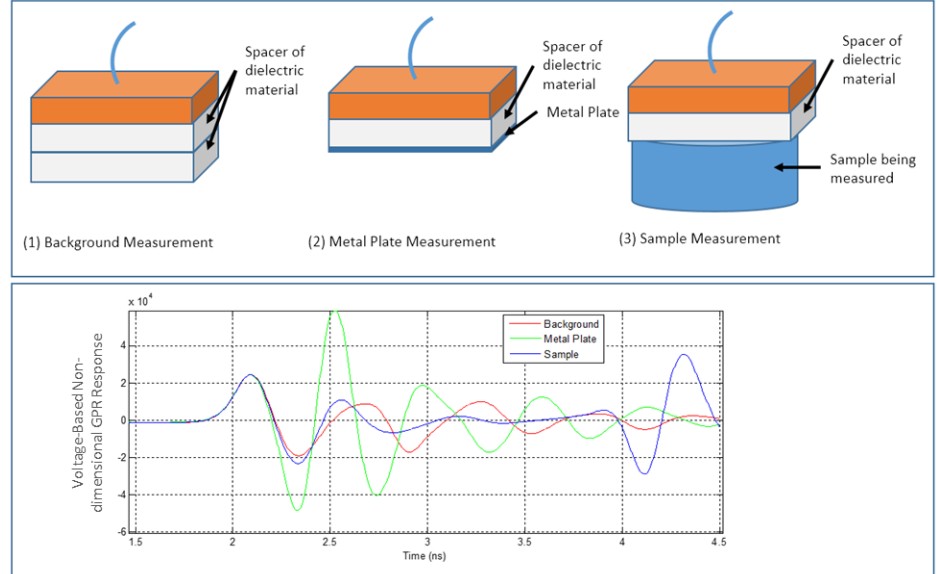

**Figure 9.** Three different measurement setups for surface dielectric (**top**) and examples of the corresponding scan data (**bottom**).

The averaged background measurement is subtracted from the metal plate and asphalt specimen measurements to isolate the reflections from the two different surfaces. Figure 10 shows the isolated reflections following background subtraction. The points of interest in these measurements are the amplitudes of the negative peaks.

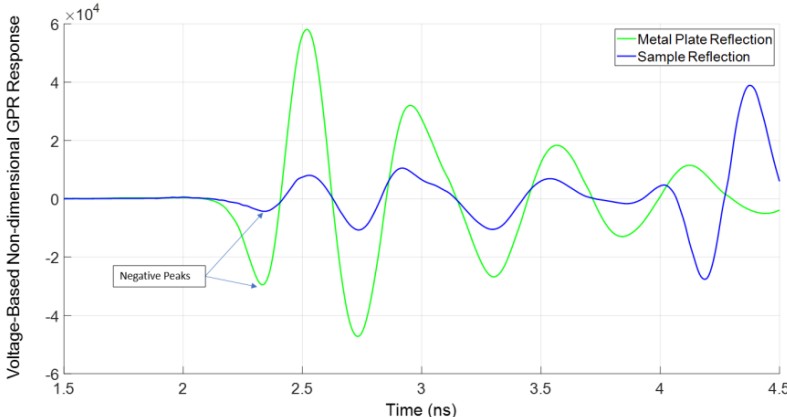

**Figure 10.** Reflection waveforms from the metal plate and sample following subtraction of the background measurement.

The surface dielectric is obtained using the reflection coefficient as an input into Fresnel's equations. The reflection coefficient ($\rho$) is calculated taking the ratio of the peak-to-peak amplitude of the asphalt specimen reflection to the amplitude of the metal plate reflection, as shown in Equation (1):

$$\rho = -\frac{A_{sn}}{A_{mn}}$$ 
(1)

Where:

$\rho$ = reflection coefficient at the interface between the spacer and the sample;
$A_{sn}$ = negative peak amplitude of sample reflection;
$A_{mn}$ = negative peak amplitude of metal plate reflection.

Fresnel's plane wave equations are used for transverse electric field polarization. Equation (2) shows this equation solved for the dielectric of the lower medium, $\epsilon_2$, which in this case would be the dielectric of the sample being tested.

$$\epsilon_2 = \epsilon_1 \sin^2 \theta_i + \epsilon_1 \left[ \frac{(1-\rho)}{(1+\rho)} \right]^2 \left(1 - \sin^2 \theta_i\right) \tag{2}$$

Where:

$\rho$ = reflection coefficient at the interface between the spacer and the sample;
$\epsilon_2$ = dielectric of the sample being measured;
$\epsilon_1$ = dielectric of the spacer between the sample and the bottom of the antenna; and
$\theta_i$ = angle of incidence, which is calculated from the known separation distance between the transmitting and receiving antennas and the thickness of the spacer.

### 2.3. Approach 2: Dielectric Obtained from Two-Way Travel Time and Known Thickness (TOF Dielectric)

Within the framework of the asphalt specimen measurements, the time-of-flight (TOF) dielectric calculation is the more robust of the two proposed dielectric calculation methods in that the evaluation is based on wave propagation through the entire depth of the evaluated asphalt specimen rather than surface only. The method is therefore less sensitive to local inhomogeneity since the response is averaged over a longer distance within the specimen, but it requires precise input of dimensions and involves more calculations to obtain the dielectric. This robustness makes it attractive for asphalt specimen dielectric measurements, but the requirement of known dimensions makes the approach inapplicable for field dielectric measurement since the thickness of the pavement is not known *a priori*. However, the resulting dielectric measured by the TOF method and the SR method is theoretically identical. A two-way travel time measurement obtained from energy propagating through the sample is achievable for a 6 in. (15 cm) diameter sample. For an asphalt specimen, the different propagation paths impacting the energy arriving at the receiving antenna are shown in Figure 11.

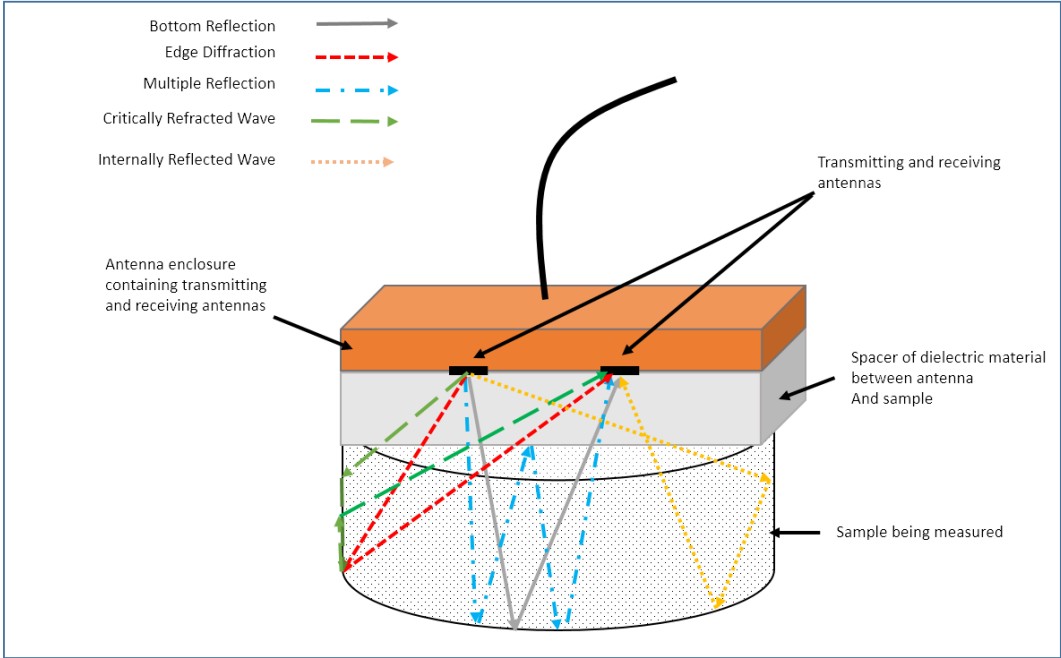

**Figure 11.** Cut-away view showing the ray paths indicating sources of major reflections and diffractions arriving from bottom and sides of the sample being measured.

The travel times of the different ray paths can be calculated versus the arrival time of the portion of the reflection of interest from the bottom of the sample being measured. Noting that the surface reflection contains a negative peak preceding the positive peak and arriving just 0.34 ns following the onset of the reflection (see Figure 6), the arrival time of this peak can also be used as the bottom reflection arrival time indicator. Choosing the initial negative peak as the common indicator of arrival (for both top and bottom TOF) is advantageous to the positive peak in that it occurs earlier in time, which is easier to isolate from subsequent reflection and diffraction arrivals. The tradeoff in using the negative peak is that it is lower in magnitude than the following positive peak. Unlike the SR method, this can be addressed by using a metal plate to amplify the signal magnitude since the TOF-dielectric method requires only the indicator of arrival, not amplitude.

In Figure 12, the arrival times of the peaks of the bottom reflection are plotted versus the calculated onsets of the reflections and diffractions involving the bottom of a 6 in. (15 cm) diameter sample. Close examination of Figure 12 shows that, at an asphalt specimen thickness range between 0.8 in. to 3.1 in. (2 cm to 8 cm), the negative peak arrival time is only preceded by the critically refracted path. For samples with a thickness less than 0.8 in (2 cm), the negative peak arrival time is preceded by the onset of the multiple reflection within the sample. At sample thicknesses greater than 3.1 in. (8 cm), the onset of the internally-reflected path precedes the negative peak from the bottom reflection.

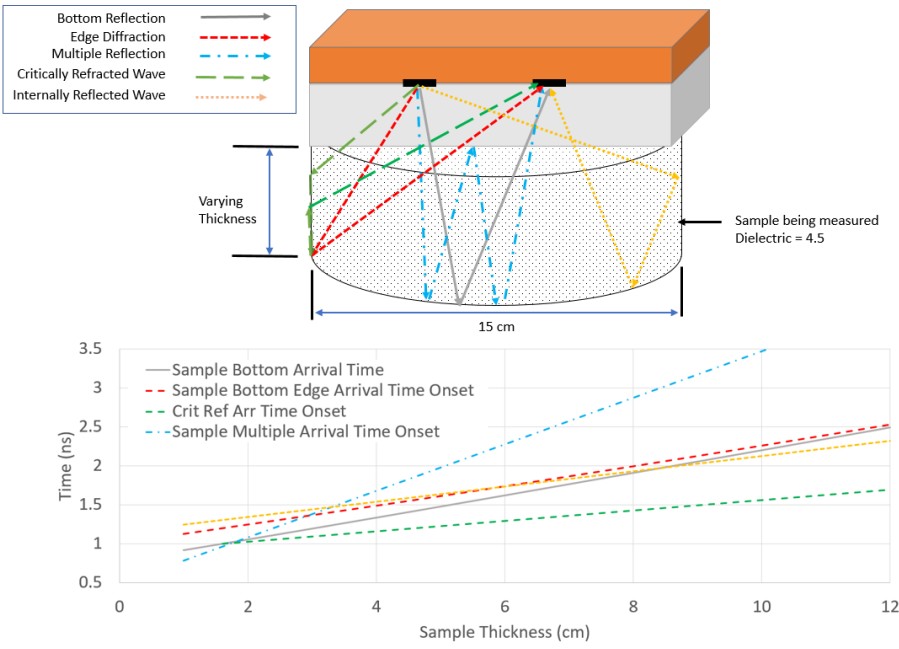

**Figure 12.** Chart of sample bottom reflection arrival times versus the onset of the arrivals from other reflection and diffraction paths for a 6 in (15 cm) diameter sample with varying thicknesses.

It is assumed that the interferences of the other travel paths are considered negligible, or removable in the analysis. It is also assumed that the increase in signal-to-noise ratio in the TOF-dielectric method by using a metal plate reflector nullifies any apparent shift in the arrival indicator that otherwise may have been significant in an asphalt to air reflection. These assumptions and their robustness were tested versus field data in repeat measurements and assessment of variable air void content asphalt specimens (discussed in the following sections). Additionally, the significance of the internally reflected wave is evaluated with a controlled laboratory experiment with 4.5 in. (11.4 cm) as well as comparison to field measurements and subsequent core ground truth for asphalt specimens with target thicknesses of 3.75 in (9.5 cm). Four measurements, shown in Figure 13, are required for the TOF dielectric measurement. The corresponding scans of data from each measurement are shown in the

bottom portion of Figure 13. Steps to remove the clutter in the scans arriving from other reflections and diffractions are also required and given herein.

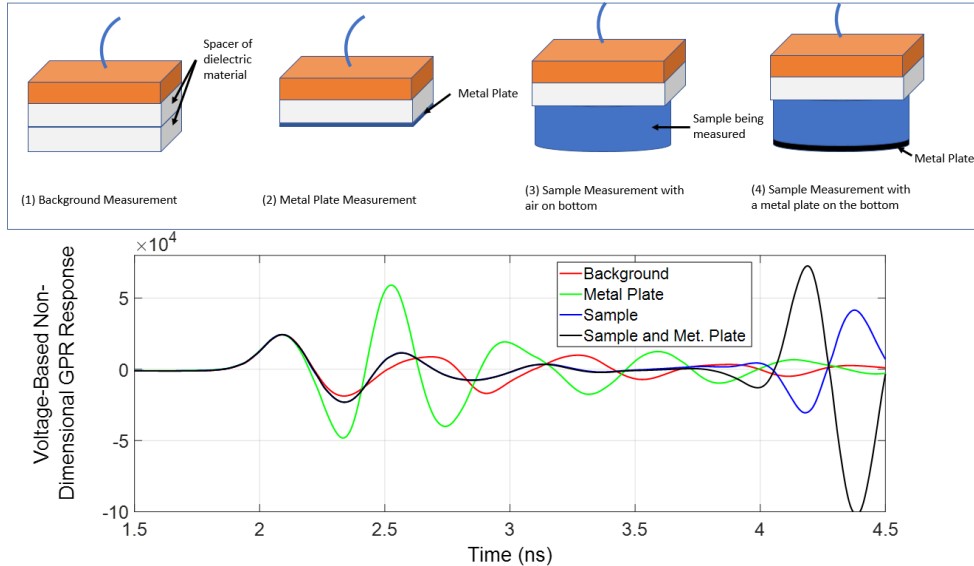

**Figure 13.** Four different measurement setups for travel-time based dielectric (**top**) and examples of the corresponding scan data (**bottom**).

The subtraction of the second measurement, with a metal plate on the bottom, from the first measurement, which was made with a second spacer, provides an isolated reflection from the bottom of the spacer. The subtraction of the third measurement from the fourth measurement provides an isolated reflection from the bottom of the sample. The two isolated reflections are shown in Figure 14. The first-arriving negative peaks of these reflections are used for the calculation of the arrival time difference. As a side note, it is interesting to observe that following the negative peak arrival, the positive peak of the reflection difference from the bottom of the sample is much larger than the positive peak of the metal plate reflection from the spacer bottom. The constructive interference of the different propagation paths shown in Figure 14 contribute to the observed amplitude.

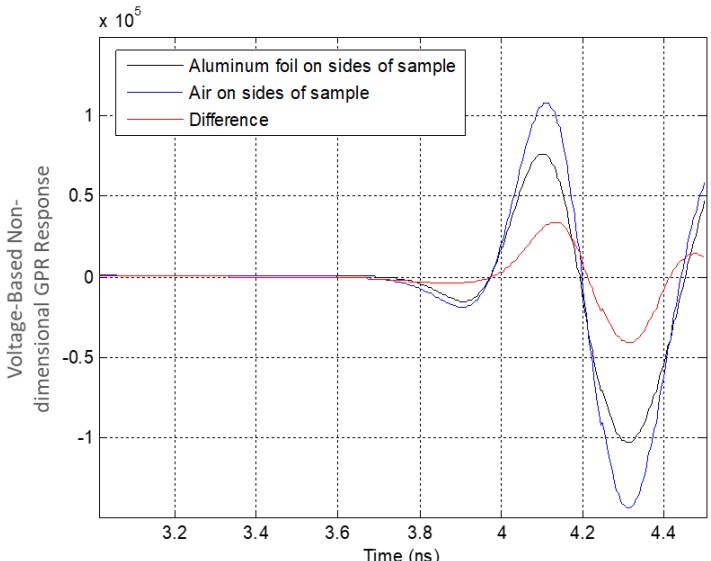

**Figure 14.** Isolated reflection from spacer bottom and the reflection difference at the sample bottom.

Given the known separation distance between the transmitting and receiving antennas, the dielectric of the spacer, the thickness of the spacer, and the thickness of the sample being tested, it is possible to calculate the dielectric of the sample using the measured travel time shown in Figure 14. This requires accommodating for the angles of refraction at the interface between the spacer and the sample. The resulting equation is a 4th order polynomial that was used to generate look-up tables for arrival times versus dielectric that were then compared to corresponding arrival times calculated using straight ray-paths. Differences between the calculated dielectrics for typical asphalt specimen dielectrics and thicknesses were in the order of 0.01 or less. For context, a change in dielectric of 0.01 corresponds to a change in air void content of less than 0.2% for the air void content versus dielectric curves generated in this study. This discrepancy is considered to be acceptable for the purpose of measuring asphalt specimen samples, since it is well within the uncertainty of the air void content measurement. Consequently, the straight ray-path calculation method is described below.

The geometry used in the calculation of the TOF dielectric is shown in Figure 15 along with the known parameters. Knowing the travel time obtained from Figure 14, the dielectric ($\varepsilon_2$) can be determined using Equations (3) through (7). The travel time of the path associated with the reflection from the top and bottom of the spacer is calculated in Equations (5) and (6). Substitution of the known parameters from Equations (3) to (6) permits the calculation of the dielectric of the sample. The method is extendable to more dielectric layers to accommodate items such as the thickness of the antenna enclosure. For the antenna used in this study, a 0.1 in. (0.3 cm) layer with 2.77 dielectric was added to the analysis to accommodate the known geometry and properties of the enclosure. The enclosure portion should be fixed to match the specific antenna used in the test, while the equations given in this study are generalized in that different spacer and asphalt specimen combinations can be experimented with based on the objective.

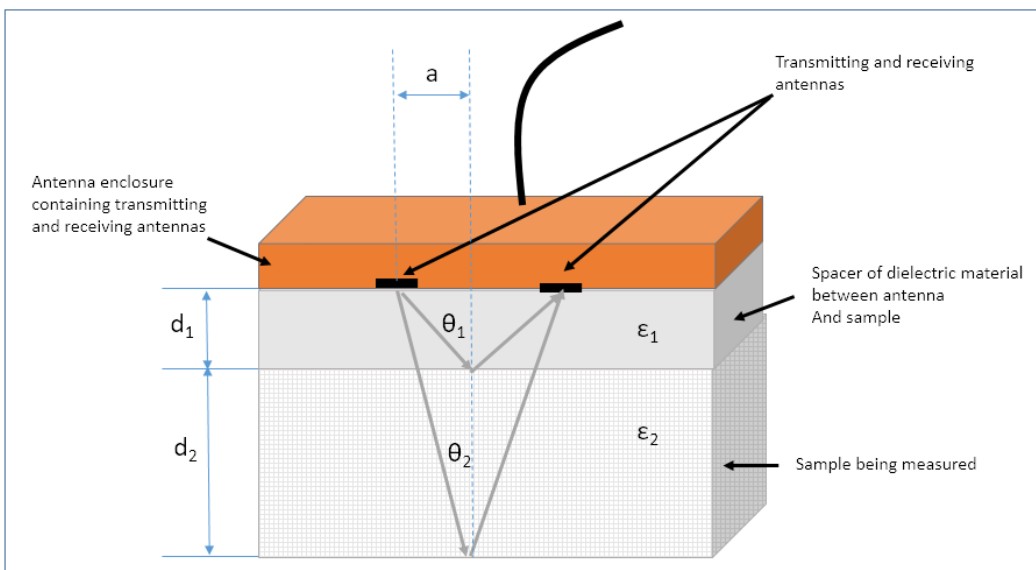

**Figure 15.** Geometry and known parameters used to calculate unknown $\varepsilon 2$ via the straight ray path assumption.

$$\theta_1 = \operatorname{atan}\left(\frac{a}{d_1}\right) \tag{3}$$

$$\theta_2 = \operatorname{atan}\left(\frac{a}{(d_1 + d_2)}\right) \tag{4}$$

$$t_1 = \frac{2d_1}{\left(\frac{c}{\sqrt{\varepsilon_1}} \cos \theta_1\right)} \tag{5}$$

$$t_2 = \frac{2d_1}{\left(\frac{c}{\sqrt{\epsilon_1}} \cos \theta_2\right)} + \frac{2d_2}{\left(\frac{c}{\sqrt{\epsilon_2}} \cos \theta_2\right)} \tag{6}$$

$$t_{meas} = t_2 - t_1 \tag{7}$$

where:

$\theta_1$ = incident angle of reflection from bottom of spacer;

$\theta_2$ = incident angle of reflection from bottom sample;

a = $\frac{1}{2}$ separation distance between transmit and receive antennas;

$d_1$ = thickness of spacer;

$d_2$ = thickness of sample;

$t_1$ = travel time of negative peak of reflection from bottom of spacer;

$\varepsilon_1$ = dielectric of spacer;

c = speed of light in air (30 cm/nanosecond);

$t_2$ = travel time of negative peak of reflection from bottom of sample;

$\varepsilon_2$ = dielectric of sample; and

$t_{meas}$ = measured travel time difference between the bottom of the sample and the bottom of the spacer.

There are a number of assumptions inherent to the application of the TOF dielectric method including the following:

1. The interference associated with the internal reflection and critically-refracted paths arriving at the same time as the negative peak of the bottom reflection is negligible;
2. Far field conditions prevail in terms of the wavefront radiated from the transmitting antenna. In other words, the intensity of the radiate energy in different directions varies only by distance.
3. The energy arriving at the receive antenna can be approximated as a plane wave.

## 3. Results

The validity of the SR dielectric and TOF dielectric methods and associated assumptions are evaluated in this section using controlled laboratory tests, followed by assessment of the predictive capabilities validated by field measurements and ground truth coring.

### 3.1. Controlled Laboratory Testing

The validity of these methods was evaluated experimentally. Comparison of SR- and TOF-based dielectric methods for repeat measurements as well as correlation with asphalt specimens with known variation in air void content was conducted. To accomplish this, production mixes from 4 days of paving on Highway 371 in Hackensack, MN were collected for the fabrication of asphalt specimens. For each of the 4 days of production, gyratory asphalt specimens were compacted to the targeted air void contents of 3%, 7%, 11%, and 15% with two replicate asphalt specimens at each air void content level. Final specimen dimensions and air void content were carefully measured using AASHTO T331. Table 1 gives all relevant information about the fabricated asphalt specimens. The asphalt specimen ID indicates the day of production, followed by the target air voids, followed by an "A" or "B" indication of the replicate.

**Table 1.** Information about Highway 371 asphalt specimens and dielectric results.

| Asphalt Specimen ID | Date | Thickness, cm | Air Voids, % | Mean, eTOF | Mean, eSR | STDev, eTOF | STDev, eSR |
|---|---|---|---|---|---|---|---|
| 1.3A | 10/1/2018 | 9.55 | 3.3% | 4.80 | 4.61 | 0.00 | 0.04 |
| 1.3B | 10/1/2018 | 9.56 | 3.3% | 4.83 | 4.55 | 0.00 | 0.11 |
| 1.7A | 10/1/2018 | 9.53 | 6.5% | 4.69 | 4.45 | 0.03 | 0.05 |
| 1.7B | 10/1/2018 | 9.54 | 6.6% | 4.69 | 4.51 | 0.03 | 0.15 |
| 1.11A | 10/1/2018 | 9.49 | 9.6% | 4.52 | 4.33 | 0.02 | 0.13 |
| 1.11B | 10/1/2018 | 9.94 | 11.7% | 4.24 | 4.42 | 0.03 | 0.04 |
| 1.15A | 10/1/2018 | 9.50 | 10.4% | 4.49 | 56.30 | 0.41 | 89.99 |
| 1.15B | 10/1/2018 | 9.49 | 10.4% | 4.43 | 4.38 | 0.03 | 0.03 |
| 2.3A | 10/2/2018 | 9.56 | 3.3% | 4.86 | 4.73 | 0.02 | 0.19 |
| 2.3B | 10/2/2018 | 9.55 | 3.3% | 4.87 | 4.66 | 0.02 | 0.04 |
| 2.7A | 10/2/2018 | 9.55 | 6.8% | 4.72 | 4.45 | 0.00 | 0.04 |
| 2.7B | 10/2/2018 | 9.54 | 6.6% | 4.73 | 4.59 | 0.00 | 0.08 |
| 2.11A | 10/2/2018 | 9.49 | 10.1% | 4.50 | 4.46 | 0.05 | 0.09 |
| 2.11B | 10/2/2018 | 9.49 | 10.1% | 4.51 | 4.44 | 0.05 | 0.03 |
| 2.15A | 10/2/2018 | 9.52 | 10.3% | 4.49 | 4.42 | 0.02 | 0.11 |
| 2.15B | 10/2/2018 | 9.51 | 10.3% | 4.51 | 4.34 | 0.02 | 0.07 |
| 3.3A | 10/4/2019 | 9.56 | 3.0% | 4.86 | 4.60 | 0.03 | 0.04 |
| 3.3B | 10/4/2019 | 9.56 | 2.9% | 4.89 | 4.57 | 0.00 | 0.01 |
| 3.7A | 10/4/2019 | 9.54 | 6.9% | 4.63 | 4.46 | 0.00 | 0.02 |
| 3.7B | 10/4/2019 | 9.54 | 6.7% | 4.64 | 4.53 | 0.03 | 0.04 |
| 3.11A | 10/4/2019 | 9.48 | 10.0% | 4.46 | 4.32 | 0.00 | 0.12 |
| 3.11B | 10/4/2019 | 9.49 | 10.0% | 4.45 | 4.25 | 0.00 | 0.14 |
| 3.15A | 10/4/2019 | 9.48 | 10.0% | 4.35 | 39.18 | 0.20 | 60.46 |
| 3.15B | 10/4/2019 | 9.44 | 9.6% | 4.46 | 4.32 | 0.00 | 0.03 |
| 4.3A | 10/6/2019 | 9.56 | 3.4% | 4.88 | 4.56 | 0.00 | 0.01 |
| 4.3B | 10/6/2019 | 9.56 | 3.2% | 4.89 | 4.53 | 0.00 | 0.07 |
| 4.7A | 10/6/2019 | 9.54 | 6.7% | 4.72 | 4.63 | 0.00 | 0.08 |
| 4.7B | 10/6/2019 | 9.55 | 6.7% | 4.66 | 4.40 | 0.05 | 0.07 |
| 4.11A | 10/6/2019 | 9.52 | 9.9% | 4.45 | 4.35 | 0.00 | 0.03 |
| 4.11B | 10/6/2019 | 9.52 | 9.8% | 4.49 | 4.32 | 0.02 | 0.04 |
| 4.15A | 10/6/2019 | 9.49 | 10.7% | 4.42 | 4.29 | 0.00 | 0.10 |
| 4.15B | 10/6/2019 | 9.43 | 10.2% | 4.47 | 4.30 | 0.02 | 0.12 |

To evaluate the precision of each method, as well as the dependence on the orientation of the asphalt specimen, each asphalt specimen was tested twice on one side, then flipped 180 degrees to test on the other side. The precision and acceptability of the SR dielectric and TOF dielectric were evaluated by taking the standard deviation of each set of three measurements of the 40 fabricated asphalt specimens shown in Table 1. It can be readily observed that asphalt specimens 1.15A and 3.15A showed the highest dielectric standard deviation in both TOF and SR dielectric measurement sets. Additionally, the measured dielectric constants (56.3 and 39.18) of the SR method are not within the range of possible values for asphalt pavement, suggesting a data quality issue. These incorrect measurements could have resulted from a number of possible issues such as improper data collection. Consequently, specimens 1.15A and 3.15A were not included in the results analysis. The results are also visualized in Figure 16, revealing that the TOF dielectric method had significantly less variability in repeated measurements of the same asphalt specimen, regardless of the surface tested. The x-axis location is determined by the specimen ID with the first digit corresponding to the date and the 10th place presentation of the specimen on the plot incrementing from 1 to 8 depending on the order it shows up in Table 1 within the given set of 8. In Figure 16, the horizontal red line indicates the AASHTO PP 98-19 defined acceptable level of 0.08 for dielectric variation in air-launched dielectric measurements [30]. The value corresponding to the red horizontal line location in Figure 16 was determined by repeated measurements on high density polyethylene (HDPE) with a known dielectric ranging from 2.30 to 2.35. This acceptable level of variation was also chosen since it typically corresponds to less than 1 percent variation in air void content, which is within the uncertainty of the air void content measurement itself. This value was used for reference in determining acceptable precision of the asphalt specimen dielectric measurements in this paper, since it is also precise enough to distinguish air void content within 1% on a typical project. Using 0.08 as the criteria, if the repeat measurement standard deviation

was less than 0.08 they were considered acceptable. Based on this criteria, thirty-eight of the 40 TOF measured dielectrics and 27 of the 40 SR dielectrics were within the 0.08 acceptable tolerance, with all remaining TOF dielectrics within 0.05. Further, the $R^2$ of dielectric vs air void content for all asphalt specimens, regardless of production day, was 0.95 and 0.69, respectively, for the TOF dielectric and SR dielectric. The superior TOF dielectric precision is presumably due to traveling throughout the entire evaluated asphalt specimen depth, as opposed to a localized surface measurement. The assumption of homogeneity is apparently more sensitive when using the SR dielectric method than when using the TOF dielectric method. This precision analysis suggests the TOF dielectric is more appropriate for use, although the surface reflection method can be a useful independent analysis of the data quality.

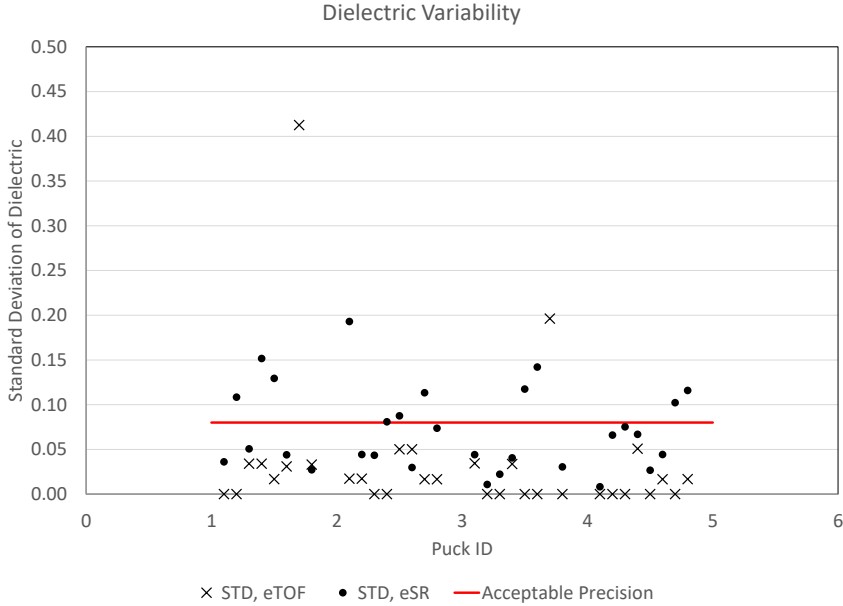

**Figure 16.** SR and time-of-flight (TOF) dielectric standard deviation versus asphalt specimen ID for all 38 evaluated asphalt specimens, where the ID is.

To evaluate the effect of edge interference on the back wall arrival time calculation in the TOF dielectric method, a test was conducted with metal indicators placed at strategic locations. The test was performed to assess the interference of the internally-reflected and critically-refracted arrivals for a typical asphalt specimen thickness. Noting that the critically-refracted propagation mode does not get excited and the polarity of the reflection from an internally reflected waveform reverses when air is replaced by metal on the outside of the sample, two measurement sequences were made on a sample 4.5 in (11.4 cm) in thickness, one with air surrounding the sides of the sample and one with aluminum foil wrapped around the sample. The isolated bottom reflection differences were obtained for each measurement sequence (i.e., blue scan in Figure 14). These two scans are plotted, together along with the difference between these two scans, in Figure 17. According to calculations, the critically refracted wave onset precedes the negative peak of the bottom reflection by about 0.5 ns. No significant energy is observed at about 3.4 ns.

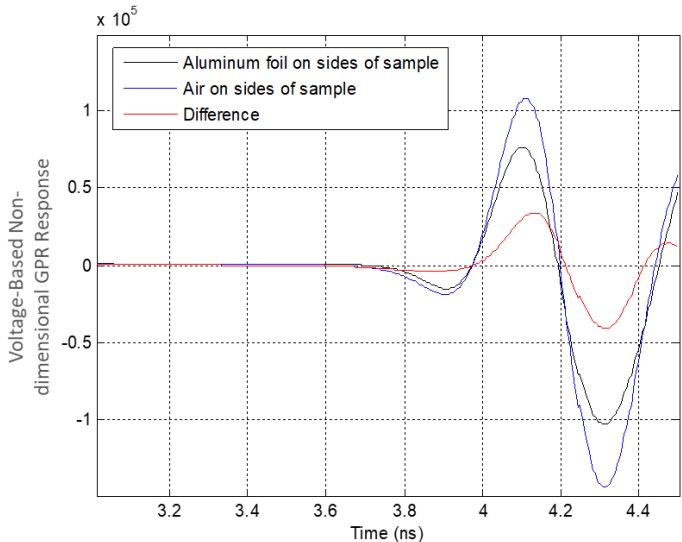

**Figure 17.** Comparison of isolated reflection differences from scans obtain with and without aluminum foil wrapped around the sides of the sample.

As anticipated, the addition of aluminum foil to the sides of the specimen destructively interfered with the bottom reflection, decreasing its amplitude due to the polarity reversal associated with the reflections from the sides of the sample. The contribution of the internal reflection from air on the sides constructively interfered with the bottom reflection. For the purposes of the TOF dielectric analysis, the change in the arrival time of the peak of the negative reflection caused by the interference from the internal reflection must be determined. Zooming in on this peak in Figure 18, it is observed that the peak arrival time changes by about 5 picoseconds. Considering that the constructive and destructive interference push the negative peak arrival time in different directions, the location of the negative peak without interference is somewhere between the two observed locations. However, this provides an idea of the impact of the internal reflections on the arrival time. A change in arrival time of 5 picoseconds translates to a change in calculated dielectric of approximately 0.03 for a sample with a 5.9 in. (15 cm) diameter, dielectric of 4.7, and a thickness of 4.5 in. (11.4 cm). The impact of the internal reflection in terms of calculating TOF dielectric is therefore not negligible, but it is manageable and in many cases insignificant.

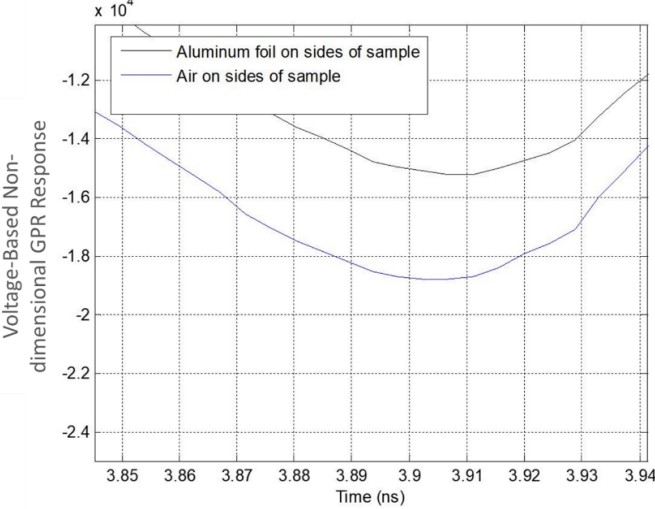

**Figure 18.** Zoomed in view of the comparison of isolated reflection differences from scans obtain with and without aluminum foil wrapped around the sides of the specimen.

### 3.2. Field Validation

For this coreless calibration method to be implemented successfully, the asphalt specimen air void content versus TOF dielectric relationship generated in the laboratory must be able to accurately predict the air void content of the placed pavement using a field air-launched dielectric measurement. To validate the accuracy of the proposed method in the field, two asphalt pavement construction projects with production mix asphalt specimens and corresponding field dielectric and core air void measurements were evaluated. Highway 371 asphalt specimens, given previously in Table 1, are compared to static air launched dielectric measurements and corresponding core air void measurements on the first and last of the 4 days of the mainline final lift placement (10/1/2018 and 10/6/2018). A similar field validation for a day of production on a construction project (Highway 60) with a significantly different mix design is also shown in Tables 2 and 3 and included in Figure 19, using a high and low field dielectric and core locations. Asphalt specimen predictions are developed primarily using the rational exponential model proposed by Hoegh and Dai et al. (HD model) and later compared to the most commonly used exponential model (conventional model) [23]. These models are given by Equations (8) and (9), respectively, as follows:

$$AV = exp\left(-B\left(D^{|\frac{1}{\varrho-C} - \frac{1}{1-C}|} - 1\right)\right) \tag{8}$$

where AV is the air void content; $\varrho$ is the measured dielectric constant; and B, C, and D are constants determined by a non-linear least-squared fit of the HD model.

$$AV = A \times exp(-b \times \varrho) \tag{9}$$

where, A and b are calibration constants of the conventional model.

Tables 2 and 3 give the asphalt specimen generated HD model and conventional model constants and $R^2$ for each production day. The highway # is the road route used for map reference. The high $R^2$ values indicate a strong correlation between dielectric and air void content when measuring asphalt specimens. This confirms that the mechanistic theory [8] and field observations [22,23] also hold for the proposed asphalt specimen-based method in terms of the statistically significant relationship between dielectric and air void content. However, for the proposed core-free method to be viable, the models must also be stable and accurate in predicting air void content based on field air-launched dielectric measurements.

**Table 2.** Asphalt specimen conventional model-generated dielectric to air void prediction coefficients and goodness of fit.

| Highway # | Date | A | b | R2 |
|---|---|---|---|---|
| 371 | 10/1/2018 | 255.7 | 1.79 | 0.99 |
| 371 | 10/2/2018 | 1230.5 | 2.11 | 0.98 |
| 371 | 10/4/2018 | 2035.9 | 2.25 | 0.98 |
| 371 | 10/6/2018 | 551.7 | 1.95 | 0.98 |
| 60 | 10/31/2018 | 364.6 | 2.15 | 0.98 |

**Table 3.** Asphalt specimen HD model-generated dielectric to air void prediction coefficients and goodness of fit.

| Highway # | Date | B | C | D | R2 |
|---|---|---|---|---|---|
| 371 | 10/1/2018 | 24.2 | 6.99 | 1.51 | 0.88 |
| 371 | 10/2/2018 | 24.18 | 6.24 | 1.26 | 0.95 |
| 371 | 10/4/2018 | 24.18 | 6.22 | 1.27 | 0.99 |
| 371 | 10/6/2018 | 24.55 | 6.56 | 1.36 | 0.97 |
| 60 | 10/31/2018 | 4.55 | 6.60 | 9.16 | 0.99 |

The measured dielectric constants and corresponding specimen air voids are shown in Figure 19. The HD model was also used to fit the data. Field cores were taken to validate the calibration curve established in the laboratory. The color representation of the cores matches the production day of the asphalt specimens. Error bars were also added to the core validation data points based on the acceptable precision range of the air launched dielectric measurement of 0.08 [30], and Minnesota Department of Transportation companion core tolerance for bulk specific gravity (Gmb) of 0.03 [24]. The tolerance based on the Gmb is presented in terms of the effect on air void content, which is calculated to be 1.2% given the 2.472 maximum specific gravity (Gmm) values for the production days tested (0.03/2.472 = 1.2%). All cores are in good agreement with the calibration curve except one, which validates the accuracy of the asphalt specimen-produced air void prediction of the field air void content on the same day of production. It should also be noted that the core air void content was measured using the AASHTO T166 method saturated surface dried (SSD) method, while the asphalt samples were measured using the AASHTO T331 method [31]. The saturated surface dry (SSD)-based AASHTO T166 method underestimates air void contents as compared to the vacuum sealed AASHTO T331 method at higher air void contents due to the presence of open or interconnecting voids [32]. This at least partially accounts for the only core showing a discrepancy, since the core was predicted to have the highest air void content, which is more susceptible to underestimation by the T166 method. The mean shift in Gmb caused by the SSD-induced bias for air void contents at a mean of 11.2% was reported by Cooley et al. [32] to be 0.041. The corresponding air void content assuming this bias is represented by solid red in Figure 19. It can be observed that the corrected core is significantly closer to the prediction, but still outside of the tolerance. Overall, the field observations validated the reasonableness of the coreless calibration predictions.

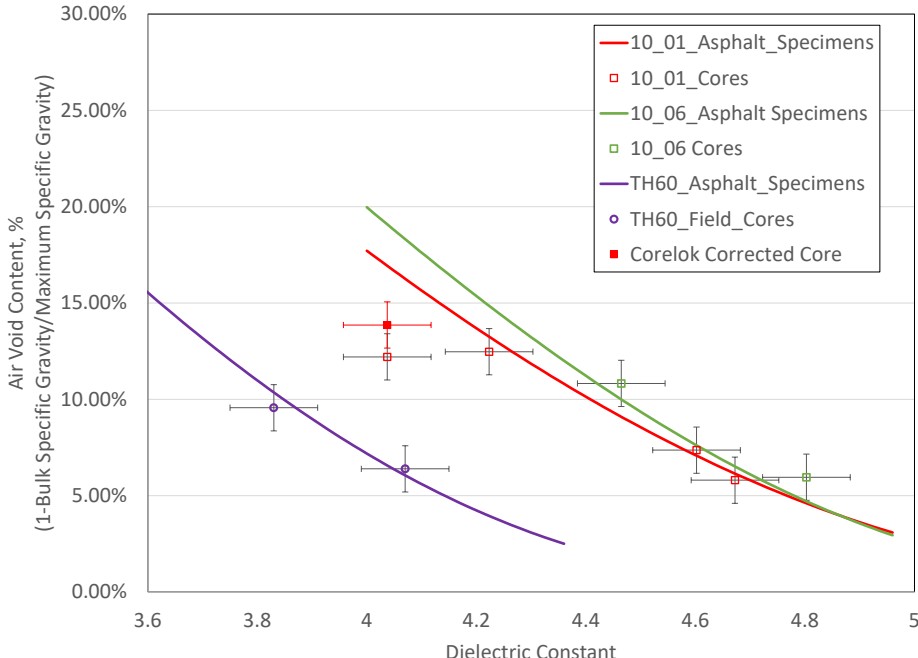

**Figure 19.** Each production day asphalt specimen generated dielectric to air void content models and corresponding field dielectric and core validation values.

While the results above validate that the same-day production mix asphalt samples give reasonable prediction of the field air void content, it may not be feasible in implementation to fabricate and measure specimens on every day of production. To assess the potential of field predictions based on limited asphalt specimen data, the stability of the models is assessed along with ability to use the first day of production asphalt specimen model to predict the last day of production field testing. The stability of the HD model and conventional model is assessed by analyzing the variation in air

void prediction by the 4 days of asphalt sample prediction curves. Since all days had the same mix design and proportions as reported by the test summary sheets, all 4 days are considered the dataset population. In this case, the 95th percentile confidence interval of the population can be calculated by the 2nd standard deviation and used to represent the stability of the models. This 95th percentile range is applied to the asphalt predictions from the first day of production in Figure 20 for both HD and conventional models, along with field cores taken on both days. It can be observed that both models are more stable at low-air-void-content, high-dielectric locations. It can also be observed that the HD model is comparatively more stable than the conventional model at higher air void content ranges, with the conventional model expanding to give unrealistic and greater 95th percentile range air void predictions. Both models give a reasonable prediction of cores corresponding to dielectric values greater than 4.2, with the HD model showing slightly greater accuracy. The core taken at a 4.04 dielectric value is extremely over-predicted when using the less stable conventional model. The HD model over-predicts the air void content, but it is within the uncertainty of the core and prediction ranges. Accounting for the likely underestimated core air voids by the SSD method (shown again in solid red), the core is within the uncertainty of the model. This analysis suggests that implementation of the coreless calibration method may be feasible, even without the need for asphalt specimen-generated cores every day of production.

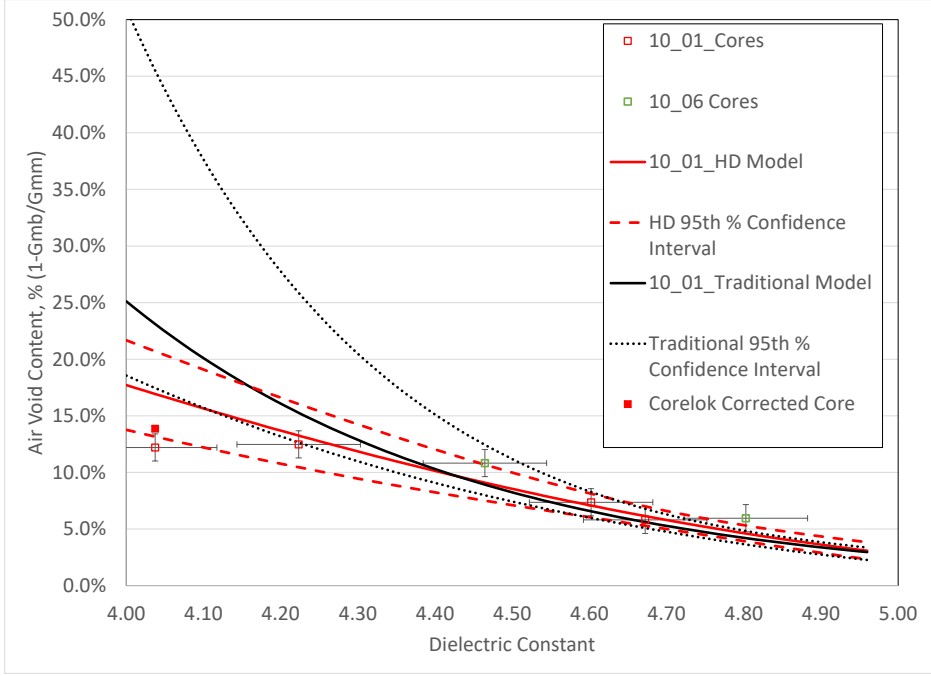

**Figure 20.** Prediction stability of Hoegh–Dai (HD) model in comparison to the conventional model when using asphalt specimen generated predictions with comparison to field measurements and core validation.

## 4. Discussion

Previous studies have shown the ability of continuous dielectric measurements to improve on the coverage and statistical significance of the conventional core sample and nuclear gauge methods. However, the requirement of field cores to calibrate the dielectric has been an impediment to widespread implementation for QC/QA. An innovative method to obtain the calibration relationship using laboratory compacted specimens is provided, thus eliminating the need for field cores. The method uses an existing laboratory gyratory compactor that is widely used in pavement industry, so no additional resources are needed. The results of the proposed method show that it is feasible to reasonably estimate field in-place air voids without taking destructive samples of the pavement.

The proposed method has been tested in detail with statistical significance under the conditions of the full-scale projects presented in this study. However, for full implementation, the method needs to be tested for a wider range of mix designs and environmental conditions. This especially includes metallic aggregates like taconite and the potential impact of magnetic aggregates on the calculated dielectric from the TOF method relative to the calculated dielectric from the SF method. The dielectric-to-air void conversion method proposed in this paper is well setup for identifying variability in asphalt mix designs and production in a wide range of field conditions. Based on fabricated samples, the proposed method can be exposed to a more detailed operator and laboratory precision evaluation as well as asphalt mix sensitivity studies since the samples are based on readily available materials and test equipment. As compared to previous destructive field sampling techniques for calibration, the findings from this paper are most impactful in moving toward the goal of continuous and accurate evaluation of asphalt pavement compaction in a fully nondestructive manner.

## 5. Conclusions

A method is for using laboratory-compacted asphalt mixture specimens to convert field-measured asphalt pavement dielectrics to air void content is proposed. Difficulties related to the small size of the specimens in relation to the DPS electromagnetic impulse signal divergence are addressed, while isolating the signal from the direct coupling noise, by introducing a plastic spacer with known thickness and dielectric properties. The precision of the approach was evaluated using repeat measurements of asphalt specimens compacted to various air void levels from two paving projects with different mix designs. The tests showed repeatability within the AASHTO tolerance for air launched dielectric measurements. Analysis of multiple days of production of the same mix design showed that the asphalt sample-generated predictions were more stable when using the HD model as opposed to the conventional model, especially at the high air void extremes. The accuracy of the proposed coreless calibration method and corresponding HD model air void predictions was validated by the reasonable agreement with asphalt pavement field air-launched dielectric measurements and corresponding ground truth cores. The results of this paper show the reasonableness of the approach and validate the coreless calibration concept in relating DPS-measured asphalt pavement dielectric to the as-built in-place air void content.

## 6. Patents

Roberts, R., 2018, U.S. Provisional Patent Application # 62/779,555;
Roberts, R., 2019, U.S. Patent Application # 16/405,209

**Author Contributions:** Each author of this paper has approved the submitted version. The authors' substantial contributions to the presented work are broken down by role as follows: conceptualization, K.H. and S.D.; methodology, R.R. and K.H.; software, R.R.; validation, K.H., S.D., and E.Z.T.; formal analysis, K.H., S.D., and E.Z.T.; investigation, K.H. and E.Z.T.; resources, R.R.; data curation, K.H. and E.Z.T.; writing—original draft preparation, K.H. and R.R.; writing—review and editing, K.H., R.R., S.D., and E.Z.T.; visualization, K.H., R.R., S.D., and E.Z.T.; supervision, S.D.; project administration, S.D.; funding acquisition, S.D. All authors have contributed substantially to the work reported.

**Funding:** This research was partially funded by the Federal Highway Administration and AASHTO under the Strategic Highway Research Program R06C (SHRP 2 R06C), funding program code M6T0.

**Acknowledgments:** The authors would like to acknowledge the contribution of Thomas Boser and Ray Betts from the Minnesota Department of Transportation (MnDOT) for their efforts in fabrication of asphalt specimens used for calibration. The authors would also like to thank Minnesota contractors and districts for accommodating data collection during live construction activities. Trevor Steiner from MnDOT was also involved in the data processing activities used in this paper. The authors would also like to thank Steve Cooper and Thomas Yu from the Federal Highway Administration for their oversight in the planning and execution of the work presented in this paper.

**Conflicts of Interest:** The authors declare no conflict of interest.

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
