# Peer review of "Toward Core-Free Pavement Compaction Evaluation: An Innovative Method Relating Asphalt Permittivity to Density"

_geosciences, doi:10.3390/geosciences9070280_

Round 1

Reviewer 1 Report

Only some very minor comments authors may wish to check for clarity.

Author Response

Thank you very much for the detailed review!  The author's point by point response is attached.

Reviewer 2 Report

The paper proposes a new practical approach for compaction evaluation. I agree that the paper is worthful for publication with a minor revision. Please consider the following comments:

-          The most frequent issue is that some units are dropped in some figures. Please make sure that Figures 3, 4, 6, 9, 10, 13, 14, 17, 18 include the units.

-          Please make sure figures has same font style, size, and color with the rest of the paper. For example, Figure 3 is inconsistent with the rest of the text.

-          Line 154, the equation is not labeled by a number, but it is introduced in a separate line. I suggest to either introduce it in the text line, or tag it with a number. For example: “following equation Fr~0.5 c (t/f)1/2, where Fr is the First…”.

-          Figure 16 was not clear to me. How acceptable line the defined? The eSR results do not have a meaningful distribution with respect to the presented red line.  It would be nice if more explanations are added to this.

The methodology and the results are acceptable, and by considering the mentioned comments the paper is eligible for publication.

Author Response

Thank you for the encouraging words and thorough review!  The point by point response is attached.

Reviewer 3 Report

This paper introduces an innovative method relating the asphalt permittivity to the density, which is a good contribution to the elimination of the need for destructive coring in assessment. The accuracy of GPR measurements have been improved by employing a plastic spacer with known dielectric properties between the specimen and antenna. The manuscript is well written. The results are good enough. Some comments are listed as follows for refinement:

1. In the abstract: 'the higher the dielectrics, the higher the density'. should be refined;

2. Does the paper mean 'dielectric constant' by 'dielectric/ dielectric value' or not? Could the indicate or specify it in the beginning in order to avoid any misunderstanding?

3. In Figure 3, the variation of dielectric constant with respect to the Scan # is shown. However, in the text, this change is described by distance/ radius. Could you provide  explanation about the relationship between the 'Scan #' and the distance?

4. In line 247, 'the time from the onset of the reflection pulse to its peak amplitude', does it mean the negative peak amplitude? Explain why it does not use the positive peak here?

5. In figure 8, it is better to write 'Dielectric=2.88' as mentioned in the text (line 264). Moreover, how does it identify the 'multiple arrival onset' from the reflection when it is overlapped with other two parameters?

6. In table 1, the Dielectric Results of 1.7 and 3.7 by SR are extremely high (56.3 and 39.18).  Explain why this anomaly exists?

7. How does it separate the other effects from air void content for your measurements, like water content and chloride content?

8. System diagram is required in line with pictures in Figures 1 and 2;

9. Mind paper format and consistent font.

Author Response

Thank you for your thorough review.  The authors believe the text is in great shape after addressing your comments.
